# A Giant Basal Cell Carcinoma of the Scalp—A Clinical Case Study

**DOI:** 10.3390/diagnostics15243114

**Published:** 2025-12-08

**Authors:** Beata Zagórska, Jacek Rutkowski, Michał Kunc, Jakub Żółkiewicz, Urszula Maińska, Michał Sobjanek, Martyna Sławińska

**Affiliations:** 1Department and Clinic of Dermatology, Venerology and Allergology, Medical University of Gdańsk, 80-210 Gdańsk, Poland; beatazagorska@gumed.edu.pl (B.Z.); ulakobus@gumed.edu.pl (U.M.); msobjanek@gumed.edu.pl (M.S.); mslawinska@gumed.edu.pl (M.S.); 2Department and Clinic of Oncology and Radiotherapy, Medical University of Gdańsk, 80-210 Gdańsk, Poland; ruten@gumed.edu.pl; 3Department of Pathomorphology, Medical University of Gdańsk, 80-210 Gdańsk, Poland; michal.kunc@gumed.edu.pl

**Keywords:** giant tumor, scalp tumor, GBCC, denial of disease

## Abstract

We describe the case of a 57-year-old female patient who reported to the dermatology outpatient clinic due to a large, ulcerated tumor of the scalp measuring approximately 12 × 10 cm. Based on the interview with the patient, it was determined that the tumor appeared more than 10 years before and was gradually increasing in size. Clinically, advanced basal cell carcinoma was suspected, which was confirmed by histopathological examination. After considering possible therapeutic options in agreement with the patient, the medical board decision qualified her for radiotherapy treatment.

**Figure 1 diagnostics-15-03114-f001:**
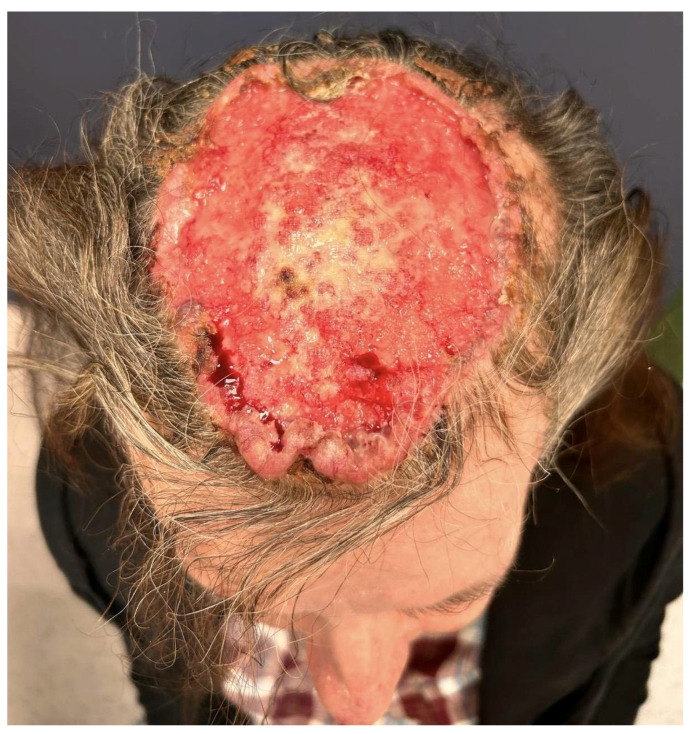
Clinical presentation. The ulcerated, bleeding tumor of the scalp measures approximately 12 × 10 cm. A 56-year-old woman presented to the dermatology outpatient clinic with an ulcerated tumor on the scalp, measuring approximately 12 × 10 cm (Figure 1). The tumor had been growing for over 10 years, and the patient had been concealing it from family members by consistently wearing a head covering. During this period, the woman consistently denied the presence of the disease, and the decision to seek medical attention was ultimately made due to pressure from her family members.

**Figure 2 diagnostics-15-03114-f002:**
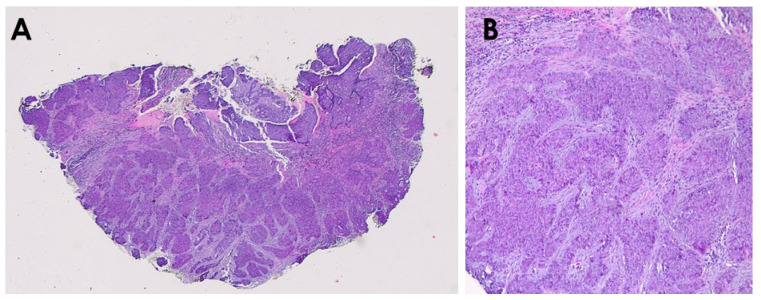
Skin biopsy revealed basal cell carcinoma (BCC) composed of anastomosing nests of basaloid cells within a myxoid stroma ((**A**)—haematoxylin & eosin, 20× magnification; (**B**)—haematoxylin & eosin, 200× magnification). The patient was referred to the Oncology and Radiotherapy Clinic for further treatment planning. A multi-site computed tomography scan revealed tumor infiltration in the cranial vault, and the clinical stage was assessed as T4N0M0. No distant metastases of the tumor were found. After discussing various therapeutic options with the patient (vismodegib pharmacotherapy, brachytherapy, external beam radiotherapy (EBRT)), it was decided to proceed with EBRT.

**Figure 3 diagnostics-15-03114-f003:**
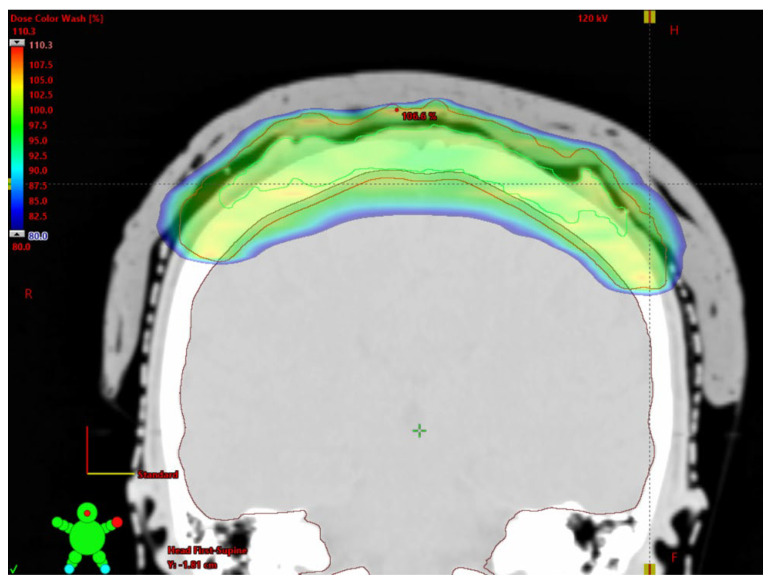
The dose distribution in irradiated area. Color wash from 80 to 110% isodose level. We delivered V-MAT (Volumetric Arc Therapy) to the mean dose in Planning Target Volume (PTV) of 66 Gy in 30 fractions using three 6 MV Flattening Filter Free arcs. Contouring on 1 mm CT slices defined gross tumor volume (GTV), clinical target volume was the GTV + 1–1.5 cm margin, and PTV = CTV + 3 mm isometric margin. Patient was immobilized with a thermoplastic mask and a patient-specific ~10 mm hydrogel bolus was used. Isocentre localization was verified with cone beam CT. Critical organ doses met all constrains; mean whole-brain dose ≤ 18%, D60Gy = 6%, D40Gy = 17%.

**Figure 4 diagnostics-15-03114-f004:**
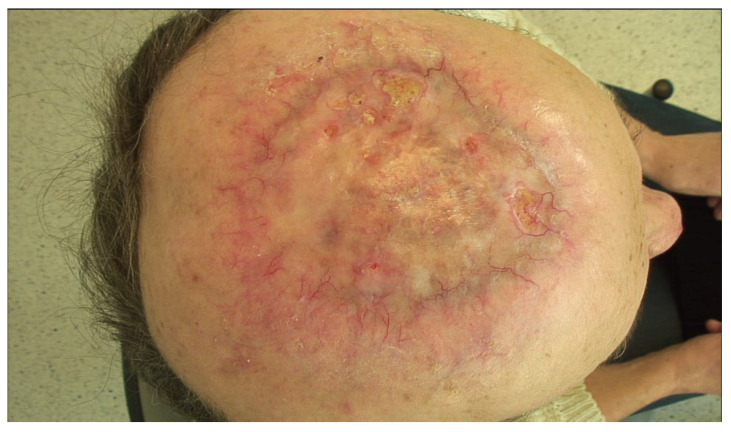
Clinical presentation 10 months after diagnosis. The treatment was well tolerated by the patient—at follow-up visits after the completion of radiotherapy, she did not report any concerning symptoms. As recommended in European guidelines, follow-up was arranged by the multidisciplinary board, with visits scheduled post-radiotherapy and subsequently every three months [1,2]. The patient remains under regular medical care. Giant basal cell carcinoma (GBCC) accounts for only 0.5–2% of all BCCs and is defined by a diameter larger than 5 cm. The most common subtype is nodular, followed by infiltrative one [3,4]. GBCC occurs twice more often in men, with a mean age of 73 years and the most common anatomical location is the trunk [3]. In addition, approximately two thirds of giant BCC cases are recurrent, with contributing factors including delayed presentation and challenges in timely diagnosis or treatment initiation [3,5]. In such cases, medical help is sought often at the insistence of family or friends [6]. Giant tumors are usually characterized by a more aggressive course, showing a tendency to invade deeper tissues including bone, muscle, and in the case of location on the scalp—periosteum, bone, dura mater and even the brain [4]. The optimal outcome of BCC therapy is complete eradication of the tumor while preserving structural and functional integrity, with consideration for the patient’s quality of life. Surgical treatment, considered the first-line method for treating typical cases of BCC, may not always be feasible for large tumors due to the potentially disfiguring effects of the procedure, medical contraindications for general anesthesia, or the patient’s refusal to undergo an extensive surgical intervention. When inoperable, irregularly shaped tumors may be effectively treated with V-MAT radiotherapy (RT). This technique is generally the superior option because it combines high conformity, deep structure sparing, reliable surface dosing with bolus, and robust image guided delivery, while electron techniques or static 3D-conformal fields are limited by geometric constraints or procedural impracticality [7]. However, when neither surgery nor RT is not feasible, systemic therapies such as hedgehog pathway inhibitors (e.g., vismodegib, sonidegib) or anti-PD-1 antibodies (e.g., cemiplimab) may be used in either palliative or neoadjuvant settings. Recent case reports have illustrated their clinical benefit in advanced, inoperable BCCs where surgery and radiotherapy were not initially viable or were deferred as part of a multimodal strategy [8]. However, technological advances in radiotherapy, including the use of three-dimensional planning and radiation intensity modulation, may also be an effective treatment option for advanced BCC when other therapeutic modalities are not feasible or the patient does not consent to them [9,10].

## Data Availability

The original contributions presented in this study are included in the article. Further inquiries can be directed to the corresponding author.

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
