# Peer review of "A Giant Basal Cell Carcinoma of the Scalp—A Clinical Case Study"

_diagnostics, 2025, doi:10.3390/diagnostics15243114_

Round 1

Reviewer 1 Report

Comments and Suggestions for Authors

1. The legand of Figure 2 needs to be concise.

2. Several case reports on the treatment of giant basal cell carcinoma with sonidegib have been published. These should be discussed in the manuscript.

Author Response

- The legand of Figure 2 needs to be concise.

Response:

Thank you for this valuable comment. We would like to note that the formatting and structure of the figure legend were guided by the journal’s instructions for the Interesting Image manuscript type. For this reason, the legend in Figure 2 is more detailed than those typically found in conventional manuscript formats.

- Several case reports on the treatment of giant basal cell carcinoma with sonidegib have been published. These should be discussed in the manuscript.

Response:

Thank you for this insightful suggestion. We have addressed this point in the revised manuscript by including a sentence that acknowledges recent case reports demonstrating the clinical benefit of systemic therapies with hedgehog pathway inhibitors and anti-PD-1 antibodies in the treatment of advanced basal cell carcinomas. This addition highlights their role in both palliative and neoadjuvant settings, particularly when surgery or radiotherapy are not initially feasible or are deferred within a multimodal treatment strategy.

Reviewer 2 Report

Comments and Suggestions for Authors

Dear authors, your case report is interesting.
Usually, with such large tumors, surgery is always the first option also for greater safety in terms of total eradication.
The structure of the manuscript should be improved, creating a paragraph in which the surgical procedures used are explained, a bit of literature should be taken up regarding the guidelines usually used, the follow up you used to monitor the tumor.
Apart from the abstract you have not created paragraphs, everything is described in a confusing way in the images, the template of each journal should be respected to give greater clarity to the topic you want to present.

Author Response

- The structure of the manuscript should be improved, creating a paragraph in which the surgical procedures used are explained, a bit of literature should be taken up regarding the guidelines usually used, the follow up you used to monitor the tumor.

Thank you for your constructive comment. We would like to clarify that no surgical procedures were performed in this case, as the patient did not consent to surgery. The follow-up schedule was determined by a multidisciplinary tumor board, in accordance with current European guidelines, which state that “in cases of difficult-to-treat or advanced BCC, follow-up should be discussed by a multidisciplinary team at a frequency dictated by each individual case.”. Based on this, follow-up was scheduled after the completion of radiotherapy and subsequently every three months.

- Apart from the abstract you have not created paragraphs, everything is described in a confusing way in the images, the template of each journal should be respected to give greater clarity to the topic you want to present.

According to the correspondence with the Editor, the structure of the article has been maintained in line with the Interesting Image format.

Reviewer 3 Report

Comments and Suggestions for Authors

Please mention the tumor size in the abstract.

Please correct grammar: The optimal outcome of BCC therapy is complete eradication of the tumor while maintaining the structural and functional integrity, considering further quality of life.

Please include a dosimetry diagram as radiotherapy planning in the area is difficult, to avoid underlining brain tissue. Is photon or electron used? What energy? Any bolus, etc. Such details will benefit the readers more.

More discussion of possible radiotherapy techniques, e.g. IMRT, electron arc, stereotactic radiotherapy with different platforms would enhance the paper.

Author Response

- Please mention the tumor size in the abstract.

Response:

Thank you for this helpful comment. We have added the tumor size to the abstract in the revised version of the manuscript.

- Please correct grammar: The optimal outcome of BCC therapy is complete eradication of the tumor while maintaining the structural and functional integrity, considering further quality of life.

Response:

Thank you for this comment. The sentence has been revised for improved grammar and clarity.

- Please include a dosimetry diagram as radiotherapy planning in the area is difficult, to avoid underlining brain tissue. Is photon or electron used? What energy? Any bolus, etc. Such details will benefit the readers more.

Response:

We thank the reviewer for this comment. Detailed data on the delivered radiotherapy can indeed be useful. We used VMAT-based radiotherapy to a mean dose in the PTV of 66 Gy in 30 fractions. Three 6 MV FFF arcs were used. Contouring was performed on 1 mm slice thickness CT scans: GTV = tumor on CT plus the marker-designated area, CTV = GTV + 1–1.5 cm margin, PTV = CTV + 3 mm. Immobilization was achieved with a thermoplastic mask. A patient-specific hydrogel bolus of approximately 10 mm thickness was applied over the area of tumor infiltration. Isocentre position was monitored using CBCT. Doses to critical organs met all internal and international standards. The mean dose to the whole brain did not exceed 18% of the prescription, D60Gy = 6%, D40Gy = 17%.

We added the synthetised sentence with RT details in the reviewed manuscript.

- More discussion of possible radiotherapy techniques, e.g. IMRT, electron arc, stereotactic radiotherapy with different platforms would enhance the paper.

Thank you for your suggestions. We added the corresponding sentenced to the reviewed manuscript.